Coming out in a harsh environment: a new genus and species for a land flatworm (Platyhelminthes: Tricladida) occurring in a ferruginous cave from the Brazilian savanna

Leal-Zanchet Ana M. zanchet@unisinos.br
Marques Alessandro Damasceno
Instituto de Pesquisas de Planárias, Universidade do Vale do Rio dos Sinos—UNISINOS , São Leopoldo , Rio Grande do Sul , Brazil
Justine Jean-Lou
Electronic publication date: 2018 Dec 4
Publication date: 2018
Volume: 6
Electronic Location ID: e6007
Received 2018 Aug 8; Accepted 2018 Oct 26
Copyright: ©2018 Leal-Zanchet and Marques
Copyright year: 2018
Copyright holder: Leal-Zanchet and Marques
License: This is an open access article distributed under the terms of the Creative Commons Attribution License, which permits unrestricted use, distribution, reproduction and adaptation in any medium and for any purpose provided that it is properly attributed. For attribution, the original author(s), title, publication source (PeerJ) and either DOI or URL of the article must be cited.
License URL: https://creativecommons.org/licenses/by/4.0/

Keywords: Geoplaninae, Land planarian, Neotropical region, Taxonomy, Subterranean fauna, Cerrado

Funding: Conselho Nacional de Desenvolvimento Científico e Tecnológico CNPq 306853/2015-9 Anglo American Brasil Carste Ciência e Ambiente The work was supported by the Conselho Nacional de Desenvolvimento Científico e Tecnológico (CNPq, Nr. 306853/2015-9), Anglo American Brasil and Carste Ciência e Ambiente. The study area belongs to Anglo American Brasil, where mining activities are planned. Biologists of Carste Ciência e Ambiente were invited to perform data collection in the study area. According to the Brazilian law, mining exploitation and other activities that may have impacts to the cave environment should be based on a report of environmental impact. Specimens collected during field sampling should be identified by specialists. In the case that new species have been detected, they should be described before procceeding with any activities.

==============================
Faunal inventories in ferruginous caves from an area belonging to the Brazilian savanna (Cerrado phytophysiognomy), on the eastern margin of the Serra do Espinhaço Plateau, in southeastern Brazil, have revealed the occurrence of land flatworms. Herein, a flatworm sampled in such subterranean environment is described as a new genus and species of the Neotropical subfamily Geoplaninae, Difroehlichia elenae gen. nov., sp. nov. The new genus shows rare features within Geoplaninae, namely sub-cylindrical body, poorly developed sub-epidermal musculature and a narrow creeping sole. Some features, such as a small body and a broad sensory margin in the anterior region of the body, as well as the fact that the holotype showed signs of recent copula, may indicate an adaptation to the subterranean environment, probably representing a troglophile. Difroehlichia elenae is characterized by an almost homogeneous dark brown pigmentation over dorsal surface and body margins, a short cylindrical pharynx, and a tubular and unforked intrabulbar prostatic vesicle, among other features. The holotype shows a secondary male copulatory organ located immediately behind the primary one, both communicating with the female atrium and gonopore canal. Since the species seems to have low abundance and restricted distribution and its type-locality is affected by mining activities, major concern arises regarding its conservation.

Introduction

Land flatworms belong to the cryptofauna, since they are photophobic and depend on the moisture of their microhabitat because they do not have water-saving adaptations (Kawaguti, 1932; Froehlich, 1955a; Winsor, Johns & Yeates, 1998). Such characteristics compel them to remain hidden during the day in humid refuges, as for example the leaf litter and under stones or fallen logs (Antunes, Leal-Zanchet & Fonseca, 2012). They are predators of other invertebrates (Froehlich, 1955a; Ogren, 1995; Prasniski & Leal-Zanchet, 2009; Boll & Leal-Zanchet, 2016; Cseh, Carbayo & Froehlich, 2017). Land triclads have restricted locomotion capacity over long distances, so there are many endemic species (Sluys, 1995).

In the Neotropical region, there have been some faunal inventories on land triclads, especially in southern South America, most of them in areas of Atlantic Forest (Leal-Zanchet & Carbayo, 2000; Castro & Leal-Zanchet, 2005; Antunes, Marques & Leal-Zanchet, 2008; Leal-Zanchet & Baptista, 2009; Baptista, Oliveira & Leal-Zanchet, 2010; Amaral et al., 2014; Negrete, Colpo & Brusa, 2014). Some specific environments, however, are almost unexplored, such as the hypogean habitats. More than 11,000 caves have been documented in Brazil, representing about only 10% of the potential number of caves in the country (Auler, Rubbioli & Brandi, 2001).

Caves are relatively stable environments regarding thermal and moisture conditions (Barr, 1968; Rocha & Galvani, 2011; Pellegrini & Ferreira, 2012), thus favouring the occurrence of land planarians. However, their diversity in such environments is almost unknown. Their occurrence in subterranean habitats may be occasional, searching for a shelter, as was probably the case of a recently described species of Geoplaninae, Pasipha ferrariaphila (Leal-Zanchet & Marques, 2018). Some species may be adapted to the cave environment, completing their life cycle both in hypogean or epigean habitats (troglophiles) or inhabiting exclusively the cave (troglobites), using it for foraging and reproduction (Barr, 1968).

Recently, a faunal inventory in ferruginous caves from an area belonging to the Brazilian savanna (Cerrado phytophysiognomy), on the eastern margin of the Serra do Espinhaço Plateau, in southeastern Brazil, indicated the occurrence of land flatworms in such subterranean environments (Leal-Zanchet & Marques, 2018). Herein we study another species of land flatworm occurring in the same region, which shows features that allow its assignment to the Neotropical subfamily Geoplaninae. However, it could not been assigned to any known genus, and thus, we provide the description of a new genus and species to accommodate this flatworm.

Material and Methods

A single flatworm was collected during the day by direct sampling in the entrance zone of a ferruginous cave (FSS-0081) in Conceição do Mato Dentro (18°56′48.1″S, 43°24′27.6″W), at an altitude of 810 m a.s.l., in the state of Minas Gerais, southeastern Brazil (Fig. 1). The specimen was fixed in 70% ethyl alcohol during field work. The preserved specimen was analysed regarding colour pattern, body shape and dimensions and then photographed under a stereomicroscope. Methods described by Rossi et al. (2015) were used for histological processing of the material and analysis of external and internal characters. The material was sectioned at intervals of 6 µm and stained with Goldner’s Masson or Haematoxylin/Eosin.

Figure 1 Type-locality of Difroehlichia elenae sp. nov., a ferruginous cave located in Conceição do Mato Dentro, state of Minas Gerais, Brazil.

The asterisk indicates the cave location; the outline indicates areas impacted by mining exploitation. Modified from Leal-Zanchet & Marques (2018).

The field work was conducted under a collection license granted by Instituto Brasileiro do Meio Ambiente e dos Recursos Naturais Renováveis—IBAMA (permit number 02015.004286/2010-49).

Type-material is deposited in the Helminthological Collection of Museu de Zoologia da Universidade de São Paulo, São Paulo, state of São Paulo, Brazil (MZUSP). The electronic version of this article in Portable Document Format (PDF) will represent a published work according to the International Commission on Zoological Nomenclature (ICZN), and hence the new names contained in the electronic version are effectively published under that Code from the electronic edition alone. This published work and the nomenclatural acts it contains have been registered in ZooBank, the online registration system for the ICZN. The ZooBank LSIDs (Life Science Identifiers) can be resolved and the associated information viewed through any standard web browser by appending the LSID to the prefix http://zoobank.org/. The LSID for this publication is: urn:lsid:zoobank.org:pub:A6418F77-34C3-413B-81E9-54296A74F2F6. The online version of this work is archived and available from the following digital repositories: PeerJ, PubMed Central and CLOCKSS.

Taxonomic description

Family Geoplanidae Stimpson, 1857	
Subfamily Geoplaninae Stimpson, 1857	

Difroehlichia Leal-Zanchet & Marques, gen. nov.

Type-species: Difroehlichia elenae sp. nov. Monotypic.

Genus diagnosis: Geoplaninae with small and slender body, sub-cylindrical in cross-section, with parallel margins; eyes mono- and trilobate, arranged along the body margins, absent on the very anterior tip; sensory margin broad; sensory pits in an irregular row on anterior body third, without contouring the tip; creeping sole narrow, less than half of body width; sub-epidermal musculature poorly developed; sub-epidermal and mesenchymal musculatures without cephalic specializations; longitudinal mesenchymal muscles absent; pharynx cylindrical; prostatic vesicle intrabulbar; male atrium short with eversible penis; ascending portion of ovovitelline ducts lateral to female atrium, joining each other dorsally to female canal or atrium; female canal dorsally flexed; female atrium obliquely disposed; adenodactyls or musculo-secretory papillae absent.

Remarks

The occurrence of dorsal testes and sub-epidermal longitudinal muscles arranged in bundles conform to the subfamily Geoplaninae, which currently has 24 genera (Carbayo et al., 2013). The new genus, however, shows poorly developed sub-epidermal musculature and a narrow creeping sole, which are rare features within Geoplaninae (Ogren & Kawakatsu, 1990).

A morphological comparative analysis suggests that among Geoplaninae, only the genus Xerapoa has such a narrow creeping sole, being even narrower, with a width corresponding to one-third of body width (Froehlich, 1955b; Ogren & Kawakatsu, 1990), whereas the creeping sole shows a width less than half of body width in Difroehlichia. In addition, Xerapoa shows sensory pits opening at the tip of small papillae, main nervous system two-chord shaped, eyes monolobate, ovaries close to the pharynx, ovovitelline ducts joining behind the female atrium and a horizontal female canal (Froehlich, 1955b; Carbayo et al., 2013). Thus, the new genus can be easily differentiated from Xerapoa by having sensory pits opening through a broad sensory margin, a broad nervous plate, eyes mono- and trilobate, ovaries close to the anterior tip, ovovitelline ducts joining dorsally to female canal or atrium and female canal dorsally flexed.

Etymology

The new genus honours the researchers Dr. Claudio Froehlich and the late Dr. Eudóxia Froehlich, who developed an extensive and renowned study on land triclads. Gender: feminine.

Difroehlichia elenae Leal-Zanchet & Marques, sp. nov.

Type material. Holotype MZUSP PL2142: leg. Carste Ciência e Ambiente, 23 November 2016, Conceição do Mato Dentro (18°56′48.1″S, 43°24′27.6″W; altitude 810 m a.s.l.), state of Minas Gerais (MG), Brazil—anterior tip: transverse sections on six slides; anterior region at the level of the ovaries: sagittal sections on five slides; pre-pharyngeal region: transverse sections on 10 slides; pharynx: sagittal sections on five slides; copulatory apparatus: sagittal sections on 6 slides; posterior region: sagittal sections on four slides.

Diagnosis

Difroehlichia elenae is characterized by an almost homogeneous dark brown pigmentation over dorsal surface and body margins; pharynx cylindrical; prostatic vesicle tubular, unforked and vertically disposed with two portions: a proximal, narrow ventral portion and a distal, globose dorsal portion; ejaculatory duct short; male atrium globose occupied by a large circular fold; male and female atria separated by a constriction; female atrium with folded walls and ample lumen, tapering to communicate with the gonopore canal; length of female atrium about 2/3 of male atrial length.

Etymology

The new species pays homage to a friend, the late Dr. Elena Diehl, who contributed to the knowledge of the ecology of ants and termites in southern Brazil.

Type-locality: Conceição do Mato Dentro, state of Minas Gerais (MG), Brazil.

Description

External features

Aspect of living specimen unknown. Body elongate with parallel margins, sub-cylindrical; anterior and posterior tips rounded (Fig. 2). Length of 13.5 mm, maximum width of 1.5 mm, and maximum height of about 0.8 mm. In relation to anterior tip, mouth at 63% of body length and gonopore at 78% of body length.

Figure 2 Difroehlichia elenae sp. nov., holotype.

(A) General dorsal view of preserved specimen. (B) Dorsal view of the anterior region of body. (C) Ventral view of the anterior region of body. The arrow indicates the anterior extremity.

Dorsal surface covered with fine, almost homogeneous dark brown pigmentation. Under the stereomicroscope, the light brown ground colour is apparent on the anterior tip and on the median dorsal region. Creeping sole grayish, bordered by dark brown lateral parts of body. Whitish sensorial margin visible in the anterior body half (Fig. 2).

Eyes absent on the very anterior tip, arranged exclusively on the body margins and almost imperceptible, even under the stereomicroscope due to the densely distributed dorsal pigment. Close to the anterior tip, eyes monolobate, with pigment cups of about 8 µm in diameter. Behind the tip, eyes become trilobate with larger pigment cups (about 15–30 µm in diameter). The eyes become sparser towards the posterior tip.

Sensory organs, epidermis and body musculatures

Anterior tip

Body elliptical in cross section just after the tip. Epidermis low (3 µm) with few and sparsely distributed rhabditogen glands and few cyanophil glands with amorphous secretion. On anterior tip, openings from erythrophil glands absent.

Creeping sole occupies about 30% of body width, showing tall, erythrophil epithelium with long and densely arranged cilia. Laterally to creeping sole, ventral epidermis similar to dorsal epidermis (Figs. 3A–3C).

Figure 3 Difroehlichia elenae sp. nov., holotype, transverse sections.

(A–C) Anterior region: general view (A) and details of the creeping sole (B) and body margins (C). (D–F) Pre-pharyngeal region: general view (D) and details of body margins (E) and creeping sole (F).

Sensory margin ventromarginally arranged and wide, occupying between 15% and 28% of body width on either side of body, with low, ciliated epithelium (Fig. 3A, 3C), receiving cyanophil glands with amorphous secretion. Sensory pits, as simple invaginations (15–30 µm deep), occur in one irregular row on either side of body in approximately the anterior third of body, without contouring the tip. Behind the tip, sensory margin gradually becomes narrower and receives openings from erythrophil glands with finely granular secretion, usually 3–4 gland necks opening close to each other.

Cutaneous musculature thicker than that in pre-pharyngeal region, especially considering its relation to body height, gradually diminishing in thickness towards anterior tip. Mesenchymal musculature, slightly thicker in cephalic region than in pre-pharyngeal region, composed of oblique fibres with various orientations, and three transverse layers: supra-intestinal, sub-intestinal and subneural (Figs. 3A–3C); thickness gradually diminishes towards anterior tip.

Pre-pharyngeal region

Creeping sole (Figs. 3D–3F) with erythrophil, tall epithelium (twice taller than the rest of the epidermis), showing an irregular height and densely arranged, long cilia. Its width corresponds to between 40% and 50% of body width. Three gland types discharge through dorsal epidermis and body margins, as well as in the lateral portions of the ventral surface: rhabditogen cells with xanthophil secretion with relatively short rhammites (about 10 µm long), cyanophil glands with amorphous secretion and sparse erythrophil glands with finely granular secretion. Sparser rhabditogen cells with smaller rhabdites open through creeping sole. Glandular margin absent (Fig. 3E).

Cutaneous musculature with usual three layers (circular, oblique and longitudinal layers), longitudinal layer thin and with small bundles (Figs. 3E–3F). Ratio of thickness of cutaneous musculature to height of body (mc:h) 3%. Ventral and dorsal musculatures with similar thickness (about 8–12 µm) at sagittal plane, similar to the epidermal height, excepting on creeping sole. Cutaneous musculature almost twice thicker than epidermis on body margins (about 18 µm).

Mesenchymal musculature (Figs. 3D–3F) poorly developed, mainly composed of two layers: (1) supra-intestinal transverse (about 2–3 fibres thick) and sub-intestinal transverse (about 3–4 fibres thick). In addition, there are oblique and dorso-ventral fibres.

Pharynx

Pharynx cylindrical and short, nearly 4% of body length, occupying almost all length of pharyngeal pouch (Fig. 4A). Pharyngeal dorsal insertion slightly posteriorly shifted, but still located in anterior third of pharyngeal pouch. Mouth in median third of pharyngeal pouch. Oesophagus short (Fig. 4A); oesophagus: pharynx ratio 12%.

Figure 4 Difroehlichia elenae sp. nov., holotype, sagittal sections.

(A) Pharynx. (B) Ovary.

Pharynx and pharyngeal lumen lined with ciliated, cuboidal epithelium with insunk nuclei. Outer pharyngeal musculature comprised of thin subepithelial layer of longitudinal muscles, followed by a thicker layer of circular fibres. Inner pharyngeal musculature comprises a thick subepithelial layer of circular fibres, followed by a thinner layer of longitudinal fibres. Outer and inner musculatures have a maximum thickness of about 15 µm, gradually becoming thinner towards pharyngeal tip. Oesophagus lined with ciliated, cuboidal to columnar epithelium, with a few insunk nuclei, and coated with a thin muscle layer (about 10 µm) comprised of circular fibres interposed with longitudinal fibres.

Reproductive organs

Testes in one irregular row on either side of body, located between intestinal branches (Fig. 3D). Testes begin posteriorly to ovaries, about 4 mm from anterior tip (30% of body length), and extend slightly posterior to pharynx. Sperm ducts lateral to ovovitelline ducts, forming spermiducal vesicles posteriorly to pharynx. Laterally to penis bulb, sperm ducts ascend, enter the muscle coat and open terminally into the proximal portion of the prostatic vesicle. Intrabulbar prostatic vesicle tubular, unforked and vertically disposed (Figs. 5, 6B). This vesicle shows two portions: proximal portion narrower and ventral; distal portion globose and dorsal (Figs. 5, 6A–6B). Ejaculatory duct short and narrow, arising from posterior region of prostatic vesicle and opening into the proximal portion of male atrium (Fig. 6C), dorsally displaced. Penis of eversible type. Male atrium short and globose with irregular contour and a large circular fold (Figs. 5, 6A). Distal region of male atrium communicates with female atrium and gonopore canal through a constriction (Figs. 5, 6A). Dorsolaterally to the gonoduct, a secondary male copulatory organ occurs (Figs. 5, 6A, 6D–6E), being displaced to the right and communicating with the distal portion of the female atrium and the gonoduct. It contains a smaller circular fold than that of the primary male atrium, receiving entally the opening of a short ejaculatory duct and a tubular, almost horizontally disposed prostatic vesicle, without anatomical differentiation between proximal and distal portions (Figs. 6D–6E). A single, incompletely developed sperm duct opens into the most proximal portion of this prostatic vesicle.

Figure 5 Difroehlichia elenae sp. nov., holotype.

Sagittal composite reconstruction of copulatory apparatus. The secondary male organ is shown in green.

Figure 6 Difroehlichia elenae sp. nov., holotype, copulatory apparatus in sagittal sections.

(A) General view. (B) Detail of the prostatic vesicle with opening of a sperm duct. (C) Opening of ejaculatory duct into male atrium. (D) Primary and secondary male organs. (E) Secondary male organs.

Epithelial lining of main and secondary prostatic vesicles ciliated and columnar, with irregular height, receiving finely granular weakly stained, erythrophil, probably mixed secretion (erythrophil core and a chromophobic peripheral part). In addition, distal portion of main and secondary prostatic vesicles receives coarse granular erythrophil secretion. Muscularis of main and secondary prostatic vesicle thick (about 6–12 µm thick), constituted of interwoven longitudinal, circular and oblique fibres (Fig. 6C). Main and secondary ejaculatory ducts lined with cuboidal to columnar epithelium, with irregular height and showing sparse cilia. Few glands with amorphous, cyanophil secretion open into these ducts. Muscle coat of ejaculatory ducts thin (about 3 µm), comprised of circular and longitudinal fibres. Main and secondary male atria lined with non-ciliated epithelium, showing microvilli and receiving finely granular, erythrophil or mixed secretion from glands with subepithelial cell bodies. Dorsal wall and proximal region of main and secondary male atria, including the circular fold, lined by columnar epithelium, with cells united to each other by their basal halves, whereas their apical cell halves are free, giving corrugated appearance to epithelium (Fig. 6C). Ventral wall and distal region of main and secondary male atria lined with cuboidal to flat epithelium. Their muscularis comprised of circular fibres followed by longitudinal fibres, thicker proximally (15–20 µm) than distally (3–5 µm). Thick muscle fibres, radial disposed, cross the dorsal wall of both main and secondary male atria.

Vitelline follicles, situated between intestinal branches, well-developed (Figs. 3D–3F, 4B). Ovaries globose, somewhat pear-shaped, measuring about 0.15 mm in both longitudinal and transversal axes. They are situated dorsally to ventral nerve plate (Fig. 4B), about 3 mm from anterior tip (22% of body length). Ovovitelline ducts emerge from lateral walls of ovaries, dorsally displaced (Fig. 4B), and run posteriorly immediately above ventral nerve plate. Distal sections of ovovitelline ducts run postero-medially lateral to female atrium and unite dorsally to female canal, forming a short common glandular ovovitelline duct (Figs. 5, 7A). Proximal portion of female atrium with an antero-dorsally directed female canal. Female atrium roughly ovoid, obliquely inclined, with folded walls and ample lumen, tapering to communicate with the gonopore canal (Figs. 5, 6A, 7A). Length of female atrium about 2/3 of male atrial length.

Figure 7 Difroehlichia elenae sp. nov., holotype, copulatory apparatus in sagittal sections.

(A) Female atrium communicating with common ovovitelline duct. (B) Detail of the lining epithelium of female atrium. Arrows indicate lacunae.

Ovovitelline ducts and common ovovitelline duct lined with cuboidal to columnar, ciliated epithelium and coated with intermingled circular and longitudinal muscle fibres (6–10 µm thick). Shell glands discharge their erythrophil, ovoid granules into the common ovovitelline duct and into the posterior sections of the ovovitelline ducts (Figs. 5, 7A). Female canal lined with columnar, ciliated epithelium, receiving abundant strongly cyanophil, amorphous secretion from cell bodies located posterior to the copulatory organs. Female atrium lined with columnar to pseudostratified, ciliated epithelium (10–30 µm), with stratified appearance in some places, reaching up to 70 µm, thus irregular in height. Some ciliated lacunae occur in this epithelium (Fig. 7A). Glands of female atrium of two types: with cyanophil, amorphous and with finely granular, erythrophil secretions. Musculature of female canal and atrium poorly developed (10–15 µm thick), composed of intermingled longitudinal and circular fibres (Figs. 5, 6D).

Common muscle coat (Figs. 5, 6A, 6C–6E), consisting of longitudinal, oblique and circular fibres, poorly developed (15–50 µm thick), thicker around ental portion of male atrium, forming penis bulb. Male and female atria with continuous muscle coat. Gonoduct large, almost straight at the sagittal plane (Figs. 5, 6A). Lining epithelium of gonoduct columnar, ciliated, receiving openings of two types of glands, one producing a finely granular erythrophil and the other an amorphous, slightly cyanophil secretion. Muscularis of gonopore canal (10–15 µm) comprised of subepithelial circular fibres and subjacent longitudinal fibres.

Remarks

The holotype is relatively well preserved, despite its direct fixation in 70% ethanol during field work (Fig. 2). This specimen shows parts of an arthropod in its intestine. Female atrium contains sperm and erythrophil, coarse granular secretion in its lumen, some sperm mixed with a slightly stained, apocrine secretion, indicating recent copula. This apocrine secretion also occurs in the gonoduct.

Ecology and distribution

The type-locality of D. elenae belongs to the Brazilian savanna (Fig. 8), in an area characterized by rocky outcrops with lateritic cover. The cave surroundings show an open vegetation composed of herbs, shrubs and trees, associated with ferruginous rocks, which is characteristic of rupestrian complexes (Rapini et al., 2008; Oliveira et al., 2018). The type-locality is located close to that of Pasipha ferrariaphila, in an area planned for mining activities (Leal-Zanchet & Marques, 2018). The sampling place is a low ferruginous cave (maximal height of 1.8 m), showing 6.3 m of horizontal projection and an area of 9 m2. It is composed of banded iron rocks (Fig. 8B) covered by a crushed lateritic cap with some quartzite fragments. The cave is located at the basis of a 4 m high vertical slope (Fig. 8A). The flatworm was sampled in the entrance zone, which corresponds to 60% of the cave area. Only two samplings were carried out in the area in June (dry season) and November 2016 (wet season), the single specimen of D. elenae being collected in the latter. Other invertebrates, such as spiders (Ctenidae and Sicariidae) and insects (Zelurus, Eidmanacris and larvae of Lampyridae), also occurred in the cave.

Figure 8 Type-locality of Difroehlichia elenae sp. nov.

(A) Entrance of the ferruginous cave that represents the type-locality, at the basis of a vertical slope in rocky outcrop in the Brazilian savanna. (B) Detail of the cave lithology, showing banded iron rocks.

Discussion

The new species described herein shows a heavily pigmented body and eyes along almost the entire body length, as presented by epigean species. However, D. elenae shows a small body and a broad sensory margin in the anterior half of the body. Such characteristics may indicate an adaptation to the subterranean environment. Adaptive features, such as hypertrophy of sensory organs and reduction of body pigmentation, as well as reduction or absence of eyes, are usually found in troglobites (Barr, 1968).

In addition, the holotype of D. elenae shows signs of recent copula, which may indicate that this flatworm is using the cave environment for reproduction, probably representing a troglophile. Land flatworms usually show low dispersion ability and may have strict ecological requirements that limit their occurrence to specific habitats, some species being affected by the conservation state of the habitat (Carbayo, Leal-Zanchet & Vieira, 2002; Baptista, Oliveira & Leal-Zanchet, 2010; Amaral et al., 2014; Negrete, Colpo & Brusa, 2014). In other faunal inventories and ecological surveys, many species of land planarians were represented by unique or few specimens (Castro & Leal-Zanchet, 2005; Antunes, Marques & Leal-Zanchet, 2008; Leal-Zanchet & Baptista, 2009; Baptista, Oliveira & Leal-Zanchet, 2010; Amaral et al., 2014; Negrete, Colpo & Brusa, 2014). Similarly, only a single specimen of D. elenae was sampled in the faunal survey in ferruginous caves in the eastern margin of the Serra do Espinhaço Plateau. Hence, it is difficult to draw firm conclusions concerning the adaptation of this species to the cave environment.

The holotype of D. elenae shows a secondary male copulatory organ immediately behind the primary one. It probably is in an initial stage of development, since it receives the opening of a single, poorly developed sperm duct. Its occurrence may be either an anomaly or a functional adaptation, constituting a populational feature or even a specific or generic characteristic, the latter occurring, for example, in some earwigs (Kamimura, 2006). The finding of other representatives of the species is also necessary to shed light on this question.

The area, where the cave is located, has been intensively sampled, but the species was collected exclusively in its type-locality (A Leal-Zanchet, 2017, unpublished data). Thus, the species seems to have a restricted distribution and, considering the mining impacts in its type-locality, a major concern for its conservation is raised.

Conclusions

The new genus described herein shows rare features within Geoplaninae, such as sub-cylindrical body, poorly developed sub-epidermal musculature and a narrow creeping sole. The holotype of the new species has a small body and a broad sensory margin in the anterior region of the body, and showed signs of recent copula. Such features may indicate an adaptation to the subterranean environment, but since the specimen shows a heavily pigmented body and eyes along almost the entire body length, it probably should be interpreted as a troglophile. Considering that the type-locality is affected by mining activities, and the species seems to have a low abundance and restricted distribution, major concern arises regarding its conservation.

Supplemental Information

Supplemental Information 1 Supplementary tables (raw data) with measurements of the holotype

Supplementary Table 1. Measurements, in mm, of the holotype of Difroehlichia elenae sp. nov.

Supplementary Table 2. Body height and cutaneous musculature in the median region of a transverse section of the pre-pharyngeal region, in micrometers, and ratio of the height of cutaneous musculature to the height of the body (mc:h index) of the holotype of Difroehlichia elenae sp. nov.

Click here for additional data file.

We acknowledge Carste Ciência e Ambiente for samplings and information about the type locality, as well as for the map and photos of the sampling place. We thank G. Iturralde for the photo in Fig. 2A and the laboratory technician L. Guterres for her help in section preparation. M.Sc. E. Benya is acknowledged for an English review of the text. Dr Marta Álvarez-Presas, Dr Fernando Carbayo and Dr. Leigh Winsor are gratefully acknowledged for their constructive comments in an early version of the manuscript.

Abbreviations used in the figures:

cmc common muscle coat

cov common glandular ovovitelline duct

cs creeping sole

di dorsal insertion of pharynx

e eyes

fa female atrium

fc female canal

go gonoduct

i intestine

lu pharyngeal lumen

med main ejaculatory duct

mma main male atrium

mo mouth

mpv main prostatic vesicle

n nerve plate

o ovary

oe oesophagus

om outer musculature of pharynx

ov ovovitelline duct

pp pharyngeal pouch

sbm sub-intestinal transversal muscles

sd sperm duct

sed secondary ejaculatory duct

sg shell glands

sm sensory margin

sma secondary male atrium

snm sub-neural transversal muscles

spm supra-intestinal transversal mesenchymal muscles

spv secondary prostatic vesicle

t testes

v vitellaria

ve ventral epidermis

vi ventral insertion of pharynx

vm ventral cutaneous musculature

Additional Information and Declarations

Competing Interests

Author Contributions

Field Study Permissions

Data Availability

New Species Registration

The work was supported in part by Anglo American Brasil and Carste Ciência e Ambiente.

Ana M. Leal-Zanchet conceived and designed the experiments, performed the experiments, analyzed the data, contributed reagents/materials/analysis tools, prepared figures and/or tables, authored or reviewed drafts of the paper, approved the final draft.

Alessandro Damasceno Marques performed the experiments, analyzed the data, prepared figures and/or tables, approved the final draft.

The following information was supplied relating to field study approvals (i.e., approving body and any reference numbers):

Samplings were approved by the Instituto Brasileiro do Meio Ambiente e dos Recursos Naturais Renováveis—IBAMA (license number 02015-004286/2010-49).

The following information was supplied regarding data availability:

The raw data (measures of the holotype) is available as a Supplemental File.

Helminthological Collection of Museu de Zoologia da Universidade de São Paulo, São Paulo, state of São Paulo, Brazil (MZUSP), type-material accession number, Holotype MZUSP PL2142.

The following information was supplied regarding the registration of a newly described species:

Publication LSID:

urn:lsid:zoobank.org:pub:A6418F77-34C3-413B-81E9-54296A74F2F6.

Genus name:

urn:lsid:zoobank.org:act:3ADC149C-5468-4088-B8FE-2DAD9EC1A3B0.

Species name:

urn:lsid:zoobank.org:act:0726E87B-805D-445A-A619-39CC27B61C30.

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
