# Peer review of "Coming out in a harsh environment: a new genus and species for a land flatworm (Platyhelminthes: Tricladida) occurring in a ferruginous cave from the Brazilian savanna"

_PeerJ, doi:10.7717/peerj.6007_

## Round 0.1 · original submission · Minor Revisions

I believe that the name of the new species should be included in the abstract - for databases, etc. I also think that the abstract should be longer.

The three reviewers have suggested a number of changes and I suggest you follow most of them.

·

Basic reporting

A new genus and species of Geoplaninae are proposed on a morphological basis. Disagreement between descriptive text and figures are main concern. Literature is well explored. Article structure is correct. English is acceptable; it is not my mother language but some errors suggest a second revision by a native English speaker. The MS is valauble for publication after revision.

Experimental design

Manuscript meets Aims and Scope of the Journal. Research question is well stated. Methods are adequately detailed.

Validity of the findings

Impact and novelty are valuable and expand knowledge of morphological and systematic diverisity of Geoplaninae. Data are robust. All figures are welcome; Fig. 5 might be improved. Conclusion is supported by the results although it includes as speculative reflection on the habits of the species. The manuscript is valuable for publication.
Some organs (eyes, sensory pits, male organs) of the species are not described in detail and should be rechecked. Even the 'eversible penis' is a diagnostic feature of the genus but there is no mention to it in the descriptive section. Text on male organs matchs only partially the figures.

Additional comments

In the manuscript the authors provide a description of a new species and genus of a Geoplaninae land planarian. I made some minor corrections to the text. I also requested clarifications on ambiguos or incomplete statements of the text.

RESULTS
Adequate but incomplete in some sections.
Telegraphic style is used inconsistently along the text, although this does not affect understanding.
Some sentences are unclear and need a revision.
It is not stated whether sensory pits are present at the anterior tip of the body (or whether it could not be checked due to the condition of the histological sections).
Diagonal dorsal mesenchymal muscle layer is not reported; might it have gone unnoticed due to the sectioning plane?
The 'eversible penis' is a diagnostic feature of the genus but there is no mention to it in the descriptive section. Text on male organs matchs only partially the figures. See detailed comments and corrections along the revised text.
Fig. 5: Instead of gray, blue (or red, green etc.) color would facilitate to observe the structures represented.

DISCUSSION
Adequate.
CONCLUSIONS
Supported by the Results. Furthermore, speculative statements on the habits of the species are presented.

I recomend publication after revision.

I attached a PDF version of the MS. If requested, I'll provide the DOCX version too.

·

Basic reporting

This article is well written, understandable and with good scientific English. The structure is correct as expected in a scientific article and conforms to PeerJ standards. The figures are correct, although the quality of the photographs of the specimen described can be improved since this is the only individual that exists.

Experimental design

The experimental design is correct in terms of histological methods and morphological description. However, given the importance of describing a new species, and even more, a new genus, the need to have molecular data of this individual should have been taken into account. Nowadays the integrative taxonomy is crucial, and especially in these cases where there is little available material, it is very important to have this type of data for a possible future identification of new individuals as well as to be able to locate the new genus in a molecular phylogeny.

Validity of the findings

no comment

Additional comments

The article: "Coming out in a harsh environment: a new genus and species of land flatworm (Platyhelminthes: Tricladida) occurring in a ferruginous cave from the Brazilian savanna" is a good contribution to the increase in the known diversity of terrestrial planarians. It also describes a new genus, something that should not be underestimated. The habitat where the specimen has been found is also peculiar, so it is endowed with characteristics important enough to be considered for publication. However, I have some concerns about whether it is ready for publication or not. The difficulty in finding terrestrial planarians is well known, especially in recondite habitats, such as caves, but the danger of describing a species and a genus from a single individual is notorious. In addition, there are no molecular data that certify the phylogenetic position of the new genus, so it is difficult, without a good morphological discussion, to phylogenetically place the new individual described. Following I write my comments on the manuscript that I think should be taken into consideration.
Introduction
- Bearing in mind that not all the readers of the article have to know the terrestrial planarians, I would dedicate a part of the introduction to describe them, talking about their most important characteristics, which later could be of help to understand the characters that are described in the results and discussion sections.
-The last sentence in the introduction section is a bit rare, please, rewrite.
Material and Methods
-More information about the external appearance of the specimen would be appreciated. Colour pattern is needed for the description. If it is not available, it must be explained in the text and the reasons why.
Results
- In the remarks of the genus, only a comparison is made with the genus Xerapoa, based on a characteristic, narrow creeping sole, which could be due to the early stage of development of the individual studied, as the authors state, or to the small size of the specimen which could respond to an immature individual. A discussion comparing the new genus with others of the group would be advisable to try to locate, at least morphologically since there is no molecular data, Difroehlichia within Geoplaninae.
- The presence of a secondary male copulatory organ is something very peculiar. Is there any other case like this in Geoplaninae? If not, have the authors considered if this individual could belong to another subfamily that is not Geoplaninae? The position of the testes (typical of Geoplaninae) could be a rare character within another subfamily that presents a cylindrical body, narrow creeping sole and other characters of this individual ...
- As a suggestion, I would place the Ecology and distribution section after Etymology, and not so towards the end of the results section. It would help to understand some characteristics, that of the way in which it is ordered now are described above.
- Has the authors or collaborators sampled in the regions near the cave to try to find more individuals? It could be possible that the cave was only a refuge for this animal and therefore does not present some typical characteristics such as depigmentation or reduction of the number of eyes.
Discussion
-A comparative discussion is NEEDED, comparing the new species with other species of the group.
- Have the authors considered the relationship of the ferruginous character of the cave and the characteristics of the animal? Maybe in the dark coloration of the specimen?

In my opinion, the article is publishable but doing a previous rethinking, and rewriting some sections before, such as the discussion. Or maybe it would be good to wait until authors have another individual available to corroborate the described characteristics and do molecular analyses. Although I understand that this may take a long time given the circumstances of the abundance of the species.

·

Basic reporting

No comment

Experimental design

The source of the geographical coordinates provided for the location, such as whether they were derived from a GPS or map, is not indicated.

Validity of the findings

No comment

Additional comments

An interesting paper that is a most welcome addition to the body of knowledge resulting from extensive research on land planarians in South America.

---

## Round 0.2 · accepted · Accept

All three reviewers agreed to accept the paper, and I agree that it should be published. Although based on a single specimen, it is a significant contribution to the taxonomy of land planarians.

# ·

Basic reporting

All the conditions required in this section are satisfactorily fulfilled.

Experimental design

Given the limitation due to the lack of available material, it is accepted that only one individual has been worked with and that there are no molecular data, hoping to have this data in the future when it is possible to find more animals of the genus and species described in this article.

Validity of the findings

No comment.

Additional comments

-The authors have added a section in the introduction about terrestrial planarians, as I suggested so that it now is more understandable for a non-planarian expert what these animals are.
-“However, it could not been assigned to any known genus, and thus, we provide the description of a new genus and species to accommodate this flatworm.” Please, change “been” by “be” in this sentence.
- All my comments have been answered satisfactorily in the responses to the reviewer. I would have appreciated some change or clarification in the text including some of those justifications.
A beautiful and interesting contribution to the increase of knowledge about the terrestrial planarians in a habitat where they have been little studied, the caves.

·

Basic reporting

No comment

Experimental design

No comment

Validity of the findings

No comment

Additional comments

In my opinion, the authors have adequately addressed the constructive points previously raised by the reviewers, by either modifying the manuscript or by clarifying their course of action in the rebuttal.